# Estimation of foveal avascular zone area from a B-scan OCT image using machine learning algorithms

**Taku Toyama** [1]*, **Ichiro Maruko**[2], **Han Peng Zhou**[1], **Miki Ikeda**[2], **Taiji Hasegawa**[2], **Tomohiro Iida**[2], **Makoto Aihara**[1], **Takashi Ueta**[1]

**1** Department of Ophthalmology, Graduate School of Medicine and Faculty of Medicine, The University of Tokyo, Tokyo, Japan, **2** Department of Ophthalmology, Tokyo Women's Medical University, Tokyo, Japan

☯ These authors contributed equally to this work.

\* toyama.ophthalmol@gmail.com

## Abstract

### Purpose

The objective of this study is to estimate the area of the Foveal Avascular Zone (FAZ) from B-scan OCT images using machine learning algorithms.

### Methods

We developed machine learning models to predict the FAZ area from OCT B-scan images of eyes without retinal vascular diseases. The study involved three models: Model 1 predicted the FAZ length from B-scan images; Model 2 estimated the FAZ area from the predicted length using 1, 3, or 5 horizontal measurements; and Model 3 converted the FAZ area from pixels to mm$^2$. The models' performance was evaluated using Mean Absolute Error (MAE), Mean Squared Error (MSE), and the Coefficient of Determination (R$^2$). The FAZ area was subsequently estimated by sequentially applying Models 1→2→3 on a new dataset.

### Results

Model 1 achieved a MAE of 2.86, MSE of 17.56, and R$^2$ of 0.87. Model 2's performance improved with the number of horizontal measurements, with the best results obtained using 5 lines (MAE: 40.36, MSE: 3129.65, R$^2$: 0.95). Model 3 achieved a MAE of 1.52e-3, MSE of 4.0e-6, and R$^2$ of 1.0. The accuracy of FAZ area estimation increased with the number of B-scan images used, with the correlation coefficient rising from 0.475 (1 line) to 0.596 (5 lines). Bland–Altman analysis showed improved agreement between predicted and actual FAZ areas with increasing B-scan images, evidenced by decreasing biases and narrower limits of agreement.

### Conclusions

This study successfully developed machine learning models capable of predicting FAZ area from OCT B-scan images. These findings demonstrate the potential for using OCT images

**Data Availability Statement:** The OCT B-scan images, corresponding FAZ measurements, and the code used in this study are publicly available on Figshare (DOI: 10.6084/m9.figshare.27262257.v1).

**Funding:** This work received funding from the Japan Society for the Promotion of Science (JSPS) KAKENHI (Grant Number: 22K16963) specifically to cover the publication fees. The funders had no role in study design, data collection and analysis, decision to publish, or preparation of the manuscript.

**Competing interests:** The authors have declared that no competing interests exist.

to predict OCTA data, particularly in populations where OCTA imaging is challenging, such as children and the elderly. Future studies could explore the developmental mechanisms of the FAZ and macula, providing new insights into retinal health across different age groups.

## Introduction

The technological advancements in optical coherence tomography (OCT) and OCT angiography (OCTA) have revolutionized our understanding of retinal physiology and pathology by providing detailed information on the neural microstructure and microvasculature. Among OCTA-derived parameters, the foveal avascular zone (FAZ) holds particular significance due to its association with various retinal pathologies. For instance, in diabetic retinopathy (DR), FAZ enlargement correlates with disease progression [1], while eyes with diabetic macular edema (DME) exhibit reduced vascular density in the deep capillary plexuses and enlarged FAZ area compared to non-DME eyes. Poor responders to anti-VEGF therapy have been reported to display larger FAZ area and increased microaneurysms [2]. In retinal vein occlusion (RVO), FAZ area inversely correlates with visual acuity (VA) [3]. Eyes affected by epiretinal membrane (ERM) typically exhibit smaller FAZ areas compared to normal eyes [4]. Importantly, a smaller preoperative FAZ area is associated with a greater degree of postoperative visual recovery [5]. Moreover, the ratio of FAZ size between eyes with ERM and unaffected fellow eyes can serve as a predictor of postoperative aniseikonia [6].

Crucially, FAZ area is linked with macular structure. Premature infants typically exhibit smaller FAZ areas alongside shallower macular depressions or foveal pits in comparison to full-term infants [7, 8]. Moreover, in normal subjects, there exists an association between foveal pit morphology and FAZ dimensions [9–11].

In this study, we employed machine learning algorithms to estimate the FAZ area using B-scan OCT images. This approach may unveil the influential role of macular morphology in determining FAZ dimensions. Furthermore, such algorithms offer potential utility given the limited availability of OCTA compared to OCT. Additionally, OCTA image acquisition is more time-consuming than OCT, posing challenges for its implementation, particularly in children and elderly individuals with limited cooperation.

This study represents the first step in a broader research initiative aimed at understanding the FAZ in populations where OCTA imaging is particularly challenging, such as children and the elderly. By successfully predicting OCTA data from OCT images, we aim to bridge the gap in FAZ assessment for these vulnerable groups. Understanding the FAZ size and its developmental mechanisms in these populations could offer new insights into macular development and retinal health, potentially influencing clinical practices in the future.

## Methods

This retrospective study was approved by the institutional review board of The University of Tokyo Hospital (#2217) and was conducted in accordance with the principles outlined in the Declaration of Helsinki. During the data extraction process on 03/10/2023, the authors had access to identifiable patient information including names and dates of birth. However, these identifiers were immediately removed after extraction to maintain confidentiality. Informed consent was waived by the ethics committee given the retrospective nature of the study and the anonymization of data post-extraction.

## Programming environment and equipment

In this study, B-scan OCT and en-face OCTA images were captured using the Cirrus 6000 (Carl Zeiss Meditec, Jena, Germany). Python version 3.8.10 was utilized as the programming language, and Google Collaboratory served as the development environment, with GPU acceleration employed to expedite machine learning computations.

## Image acquisition and preparation

OCT and OCTA images of eyes without vitreoretinal diseases were retrieved from patients visiting the ophthalmology service of The University of Tokyo Hospital. Serial horizontal B-scans of the macula (6 x 6 mm) and the corresponding 6 x 6 mm superficial en-face OCTA images were utilized (Fig 1A and 1B).

The FAZ area was measured using the superficial capillary plexus (SCP) slab of the en-face OCTA images. The SCP FAZ measurement is widely used in clinical practice and research and is associated with various retinal pathologies and developmental conditions [12–14]. Moreover, studies have utilized SCP FAZ measurements in pediatric populations, such as children with amblyopia [15], highlighting its significance in our target populations. While measuring

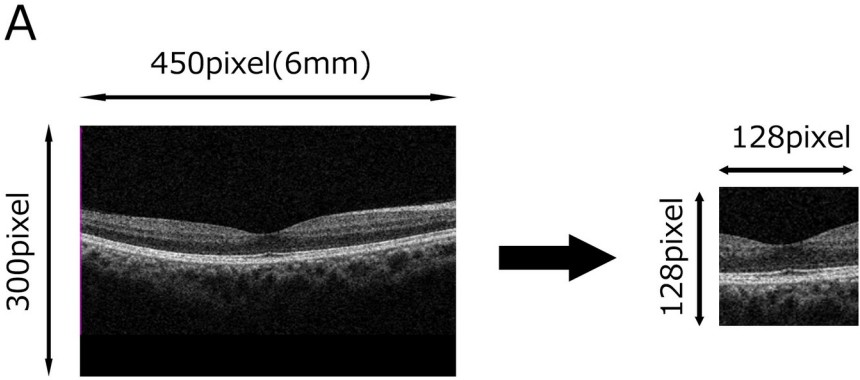

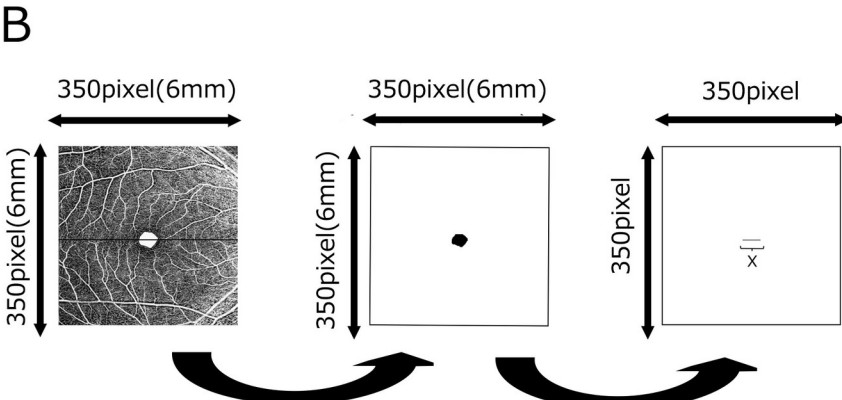

**Fig 1. Preparation of B-scan OCT and en-face OCTA images.** (A) A 128x128 pixel area centered on the fovea was extracted from a horizontal B-scan image (450 x 300 pixel) passing through the foveal center. (B) The superficial area of the FAZ is automatically calculated from en-face OCTA images (350 x 350 pixel) using internal software. The shape and size of the FAZ were extracted, and the length of the FAZ (X) passing through the foveal center is measured.

the FAZ in the full retinal slab can yield more comprehensive results in some cases [16, 17], we selected the SCP slab to align with standard clinical protocols and facilitate comparison with existing literature. Future studies will explore the application of our models to full retinal slab measurements.

A horizontal B-scan image passing through the fovea (450 pixel width x 300 pixel height) was selected and cropped to 128x128 pixels (equivalent to 1.7 mm width) with the fovea centered (Fig 1A). The internal software automatically detected and calculated the area of superficial FAZ based on the en-face OCTA image (350 pixel x 350 pixel, Fig 1B). The FAZ area was extracted, and the width of the FAZ corresponding to the B-scan OCT through the fovea was measured (designated as "X" in Fig 1B). Regarding image quality, a signal strength of 6 or higher was recommended for Cirrus 6000 [18] in a previous study. Here, images with a signal strength of 9 or higher were utilized.

## Machine learning models

Estimation of FAZ area based on single or multiple horizontal B-scan OCT images was conducted through the following Model 1–3.

**Model 1.**  A convolutional neural network (CNN) was formulated to predict the length of the FAZ segment (X) from the B-scan image (Fig 2A). A dataset comprising 1146 pairs of B-scan images and corresponding FAZ segment length was assembled for training purposes. TensorFlow and Keras were utilized for model implementation, incorporating early stopping, and model checkpoint callbacks to counteract overfitting. Additional code details can be found in S1 Code.

**Model 2.**  Neural network models were developed to predict FAZ area (Y) from its width (X) derived from a B-scan image. A dataset consisting of 475 pairs of FAZ length and area were prepared for machine learning purposes. Three variations of Model 2 were explored: Model 2 (1line) utilized a single horizontal FAZ width through the fovea, Model 2 (3 lines) incorporated three horizontal widths through the fovea and its 6 pixels upper and lower positions, and Model 2 (5 lines) integrated five horizontal widths through the fovea and its 6 and 12 pixels upper and lower positions (Fig 2B). The model architecture featured a three-layer neural network. Model performance was assessed using both training and test datasets, with the application of early stopping and model checkpoint techniques to mitigate overfitting. Detailed specifications of Model 2 and the corresponding machine learning code are provided in S2 Code.

**Model 3.**  A machine learning model was developed to estimate FAZ area in $mm^2$ (Z) from the area in pixels (Y) within a 350x350 pixel image (Fig 2C). The dataset comprised 499 pairs of FAZ area in $mm^2$ (Z) and the corresponding area presented in pixels (Y). Following data normalization, the model was constructed using a three-layer neural network architecture. Evaluation of model performance was conducted using both training and test datasets, with the implementation of early stopping and model checkpoint techniques to mitigate overfitting. The structure of Model 3 and the accompanying machine learning code are elucidated in S3 Code.

The outcomes derived from applying Model 1 → Model 2 (1 line) → Model 3 in sequentially were denoted as Estimate (1 line). Similarly, the results obtained from the sequence of Model 1 → Model 2 (3 lines) → Model 3 were labeled Estimate (3 lines), while those generated from Model 1 → Model 2 (5 lines) → Model 3 were referred to as Estimate (5 lines).

## Model evaluation

The performance of each model was assessed using the following metrics:

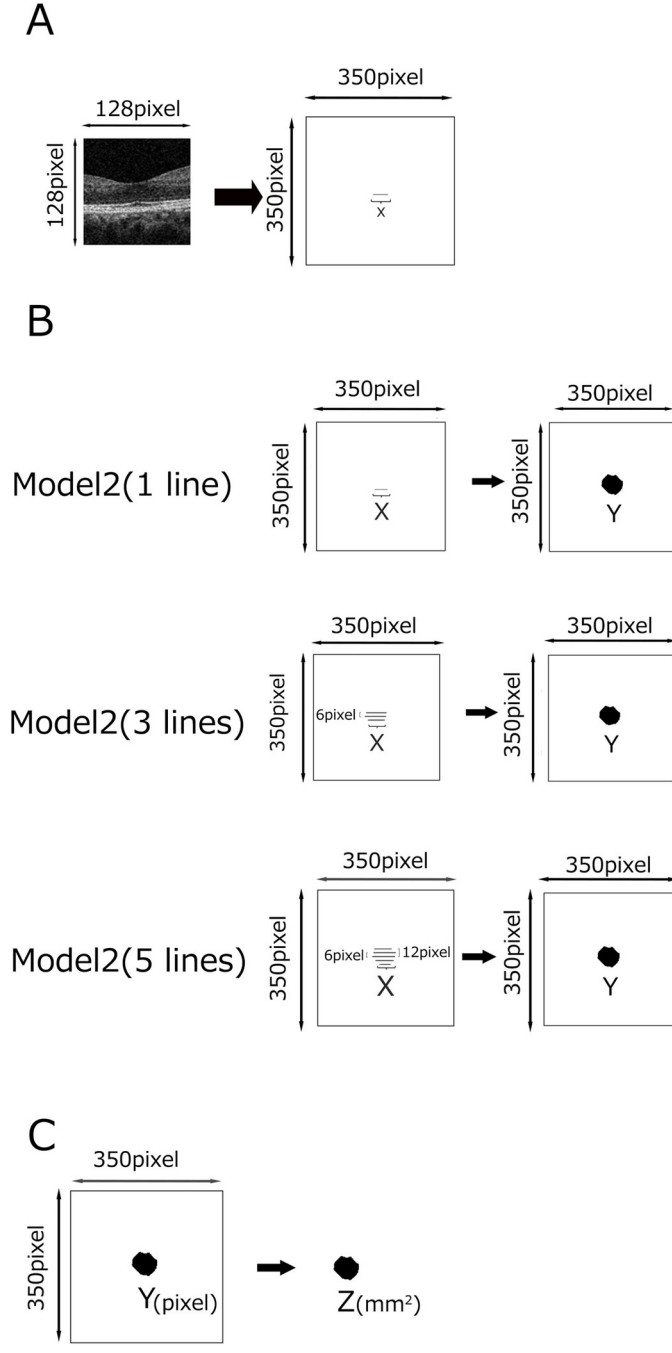

**Fig 2. Three models to estimate FAZ area from B-scan OCT images.** (A) Model 1 estimates a FAZ length (X) from a B-scan OCT image. (B) Model 2 predicts the FAZ area (Y, by pixel) based on the length (X) determined by Model 1. In this study, we increased the number of length (X) to 1, 3, and 5 to investigate whether the accuracy of the machine learning model improves. (C) Model 3 calculates Z ($mm^2$) from the Y (pixels) determined by Model 2.

- Mean Absolute Error (MAE)

- Mean Squared Error (MSE)

- Coefficient of Determination ($R^2$)

These metrics were calculated on the test dataset to assess the predictive accuracy and generalizability of the models. MAE and MSE evaluate how well the model's ability to minimize errors, while $R^2$ indicates how well the model fits the data and is employed to gauge the accuracy of regression models [19].

The performance evaluation of the developed machine learning model was carried out using a newly assembled dataset comprising 328 eyes. During the evaluation process, the model's precision and accuracy were evaluated by comparing the absolute error between the predicted FAZ area by the model and the actual FAZ area. The predictive performance of the model was compared based on estimates using 1, 3, and 5 B-scan OCT images.

## Data sharing

The OCT B-scan images and corresponding FAZ measurements used in this study are publicly available on Figshare (DOI: 10.6084/m9.figshare.27262257). All data have been anonymized to protect patient privacy, in accordance with the ethical guidelines approved by the Institutional Review Board of The University of Tokyo Hospital.

**Statistical analysis.** Pearson's correlation coefficient was utilized to assess the correlation between predicted and true values of FAZ area. Bland–Altman analyses were conducted to evaluate the agreement between predicted and actual FAZ area measurements for each estimation method [20]. To compare the estimation error among the three Estimates, repeated measures ANOVA was conducted, followed by paired t-tests for between-group comparisons. As a correction for multiple comparisons, Bonferroni correction was applied. Estimation error was calculated using the absolute difference between the estimated values and the actual values. Statistical analysis was performed using Python version 3.8.10, with a significance level set at less than 0.05.

## Results

Table 1 presents the MAE, MSE, and $R^2$ for each Model used in this study. The mean ± SD of true X values in the dataset for machine learning was 25.82±11.96. The validation data exhibited MAE, MSE, and $R^2$ values of 2.86, 17.56, and 0.87 respectively.

The mean ± SD of true Y values in the dataset for machine learning was 913.11±318.23. For Model 2 (1 line), the MAE and MSE were 105.161 and 17881.80, respectively, with an $R^2$ of 0.80. For Model 2 (3 lines), the corresponding values were 74.83, 9505.82, and 0.87, respectively. For Model 2 (5 lines), the values were 40.36, 3129.65, and 0.95, respectively.

The mean ± SD of Z in the machine learning dataset was 0.26±0.093. In Model 3, the validation data exhibited a MAE of 1.52e-3, an MSE of 4.0e-6, and an $R^2$ of 1.0 (Table 1).

Saliency maps, as described in previous studies [21, 22], were utilized to reveal the specific regions of interest within the B-scan image that the machine learning algorithm focused on during training in Model 1. Notably, these maps highlighted areas surrounding the fovea as the primary points of emphasis (Fig 3).

A new test dataset comprising 328 eyes was established to assess the precision and accuracy of predicting FAZ area using Models 1→ 2→ 3. When employing a single B-scan OCT image, the correlation coefficient (R) between the true and predicted FAZ area was R = 0.475 (P < 0.001) for the Estimate (1 line) (Fig 4A). Increasing the number of B-scans improved FAZ area prediction precision, resulting in correlation coefficient of R = 0.533 (P < 0.001) for the Estimate (3 lines) (Fig 4B), and R = 0.596 (P < 0.001) for the Estimate (5 lines) (Fig 4C).

Furthermore, to evaluate the accuracy of the FAZ estimations, the estimation error (i.e., absolute difference between true and estimated FAZ area) was calculated for each method and compared among the three estimation techniques using varying numbers of B-scan OCT

**Table 1. Performance comparison of Model 1, Model 2 (Variations 1, 3, 5), and Model 3.**

| Model | Data Type | MAE | MSE | $R^2$ |
|---|---|---|---|---|
| Model 1 | Validation | 2.86 | 17.56 | 0.87 |
|  | Test | 2.69 | 13.11 | 0.92 |
| Model 2–1 | Validation | 105.161 | 17881.8 | 0.80 |
|  | Test | 112.48 | 20994.23 | 0.75 |
| Model 2–3 | Validation | 74.83 | 9505.82 | 0.87 |
|  | Test | 82.76 | 16888.61 | 0.74 |
| Model 2–5 | Validation | 40.36 | 3129.65 | 0.95 |
|  | Test | 40.44 | 5231.16 | 0.93 |
| Model 3 | Validation | 0.00152 | 4.00E-06 | 1.0 |
|  | Test | 0.00146 | 4.00E-06 | 1.0 |

images. Repeated measures ANOVA revealed significant differences in estimation accuracy among the methods (P < 0.001). Subsequent paired t-tests with Bonferroni correction for multiple comparisons consistently demonstrated improved accuracy with an increasing number of

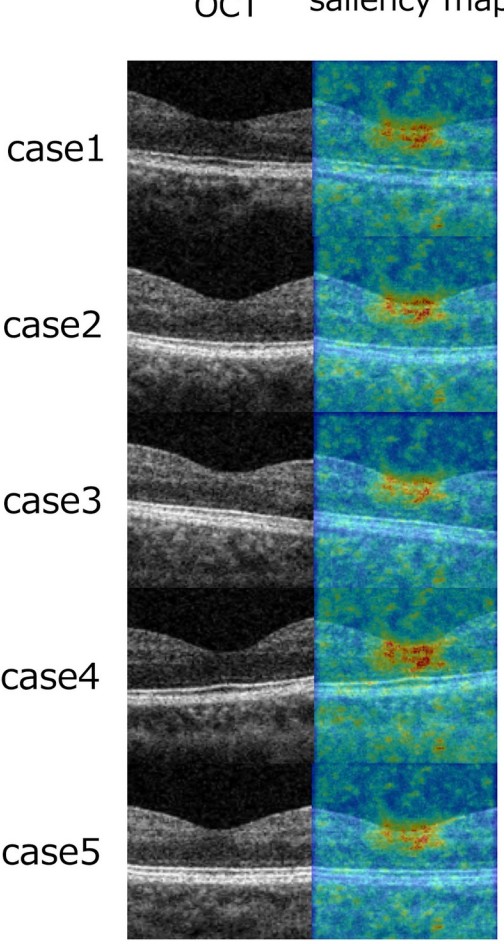

**Fig 3. Saliency maps were employed in order to elucidate which part of the B-scan images the machine learning focused on during training in Model 1.** As a result, it was evident that area around the fovea were predominantly highlighted.

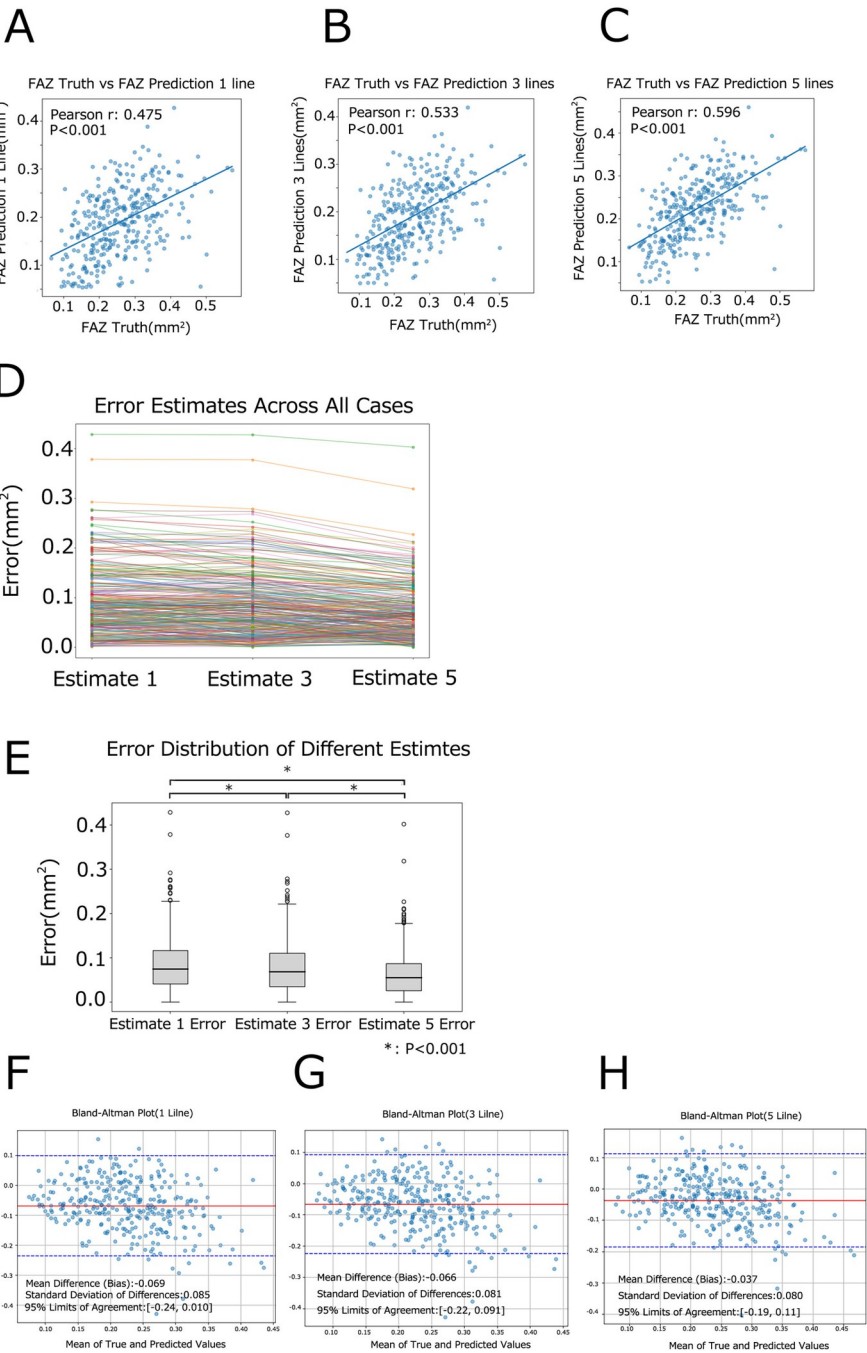

**Fig 4. Accuracy of Models 1–3 in predicting FAZ area from B-scan OCT images evaluated on the test dataset.**
Correlation coefficient (R) between actual and predicted FAZ area based on (A) 1 line, (B) 3 lines, and (C) 5 lines of B-scan OCT images. (D) The relationship between the number of B-scan OCT images (x-axis) and absolute error (y-axis) is presented for each case in the test dataset. (E) The absolute error was compared among Estimate-1, Estimate-3, and Estimate-5. P<0.001 by paired t-tests with Bonferroni correction for multiple comparisons. (F–H) Bland–Altman plots showing the agreement between predicted and actual FAZ area measurements for (F) Estimate (1 line), (G) Estimate (3 lines), and (H) Estimate (5 lines). The solid lines represent the mean differences (biases), and the dashed lines represent the 95% limits of agreement.

horizontal OCT images: from 1 to 3 ($p < 0.001$), and from 3 to 5 ($p < 0.001$). Hence, estimation based on 5 horizontal OCT images exhibited significantly higher accuracy than those based on fewer images, underscoring a clear trend of enhanced accuracy with the addition of more horizontal OCT images (Fig 4D and 4E).

To further assess the agreement between the predicted and actual FAZ area measurements, Bland–Altman analyses were conducted for the estimates based on 1, 3, and 5 B-scan OCT images. The mean differences (biases), standard deviations (SDs), and 95% limits of agreement were calculated for each case. For Estimate (1 line), the mean difference (bias) was -0.069 $mm^2$, with a standard deviation of 0.085 $mm^2$, resulting in 95% limits of agreement from -0.24 $mm^2$ to 0.10 $mm^2$ (Fig 4F). For Estimate (3 lines), the mean difference was -0.066 $mm^2$, with a standard deviation of 0.081 $mm^2$, and 95% limits of agreement from -0.22 $mm^2$ to 0.091 $mm^2$ (Fig 4G). For Estimate (5 lines), the mean difference was -0.037 $mm^2$, with a standard deviation of 0.080 $mm^2$, and 95% limits of agreement from -0.19 $mm^2$ to 0.11 $mm^2$ (Fig 4H).

These results indicate that as the number of B-scan OCT images increases, the agreement between the predicted and actual FAZ area measurements improves, as reflected by the decreasing biases and narrower limits of agreement.

## Discussion

The current study, employing three distinct machine learning models, provides a robust demonstration of the feasibility of estimating FAZ area from B-scan OCT images with a notable degree of accuracy. In alignment with prior investigations elucidating the correlations between foveal morphology and FAZ [9–11], our machine learning models effectively emphasized the fovea as a focal point for predicting FAZ area, as underscored by the saliency maps.

The comparative analysis reveals that Model 2 displays somewhat inferior accuracy, primarily evidenced by its elevated MSE values, particularly when leveraging only a single B-scan OCT image. This discrepancy can be attributed to the inherent complexity of FAZ morphology, wherein different values of Y correspond to the same X. For instance, within the training dataset, the prevalence of X = 33 underscores this variability, with corresponding Y values ranging from 574 to 1006, averaging at 799. This diversity in FAZ shape underscores the challenge of predicting the comprehensive FAZ size from a solitary horizontal line.

Notably, augmenting the number of X inputs into Model 2 resulted in significant improvements across key metrics such as MAE, MSE, and R2, underscoring the importance of incorporating multiple horizontal lines for enhanced accuracy. Despite attempts to integrate additional variables such as gender and age into the B-scan images, these efforts did not yield discernible improvements in predicting FAZ area. However, further exploration and comprehension of factors influencing FAZ area hold promise for enhancing the precision of our models in subsequent iterations.

In regard to Model 3, it employs a methodology wherein FAZ area estimation is achieved by quantifying the number of black pixels within a single 350x350 pixel image. While the precise algorithms utilized within the OCT machine remain undisclosed, the model's extremely low values of MAE and MSE with $R^2$ value of 1.0 suggests a close approximation to the internal calculation process of the Cirrus 6000.

We conducted an evaluation of Model 1, 2, and 3 using a fresh test dataset comprising 328 eyes. The MAE, MSE, and R2 values obtained were consistent with those derived from the validation dataset, indicating robust performance across different datasets. Significantly, the correlation between the true and predicted FAZ area was found to be statistically significant, underscoring the models' ability to accurately estimate FAZ area. Moreover, the incorporation of additional B-scan OCT images resulted in a notable enhancement in accuracy, as evidenced

by the progressively increasing correlation coefficients (R = 0.475, 0.533, and 0.596 for 1, 3, and 5 OCT images, respectively). This improvement in accuracy with the augmentation of OCT images was further substantiated through rigorous statistical analyses, including repeated measures ANOVA and paired T-tests with Bonferroni correction, as depicted in Fig 4D and 4E. The Bland–Altman analyses further supported these findings, showing decreased mean differences (biases) and narrower 95% limits of agreement as the number of B-scan images increased. Specifically, the bias improved from -0.069 mm$^2$ for the 1-line estimate to -0.037 mm$^2$ for the 5-line estimate, indicating reduced systematic error. The narrowing of the limits of agreement suggests enhanced consistency and reliability in the FAZ area predictions with more B-scan images (Fig 4F–4H). Such findings underscore the efficacy and reliability of our models in FAZ area prediction, affirming their potential clinical relevance and utility.

A significant limitation of our study is the inclusion of only healthy adult subjects without retinal vascular diseases. This focus may restrict the applicability of our machine learning models to populations with retinal pathologies. Previous research has demonstrated that in diseased eyes, even the built-in software of OCTA devices can be inaccurate in delineating the FAZ area [23]. Therefore, our models, trained on data from healthy eyes, may not generalize well to eyes with retinal diseases.

Additionally, our models were developed using data from adult eyes, which may limit their applicability to younger populations, including children and preterm infants. Differences in ocular characteristics and developmental stages between adults and pediatric patients could potentially affect the models' performance and generalizability in these populations.

Furthermore, the application of our models to images captured by different OCT machines may require adjustments, such as transfer learning. Variations in imaging modalities, resolutions, and proprietary algorithms among different OCT devices can influence the models' ability to accurately predict FAZ area.

Therefore, further research and validation studies are warranted to assess the performance of our models across diverse age groups, disease states, and OCT platforms, ensuring their robustness and reliability in various clinical settings.

In conclusion, our study successfully developed machine learning models capable of estimating the FAZ area from OCT B-scan images in eyes without retinal vascular diseases. We demonstrated that increasing the number of B-scan images used in the estimation process improves the predictive performance of the models. These findings suggest that OCT B-scan images could potentially be used to estimate FAZ area in situations where OCTA imaging is not feasible due to patient cooperation issues, motion artifacts, or equipment limitations. However, further studies are needed to validate the models in broader populations, including patients with retinal diseases and different age groups.

## Supporting information

**S1 Code. Python code for Model 1 used to predict FAZ length from B-scan OCT images.**
(PY)

**S2 Code. Python code for Model 2 used to predict FAZ area from FAZ length measurements.**
(PY)

**S3 Code. Python code for Model 3 used to convert FAZ area from pixels to mm$^2$.**
(PY)

## Author Contributions

**Conceptualization:** Taku Toyama, Ichiro Maruko, Han Peng Zhou, Miki Ikeda, Taiji Hasegawa, Tomohiro Iida, Takashi Ueta.

**Data curation:** Taku Toyama.

**Investigation:** Taku Toyama, Ichiro Maruko, Takashi Ueta.

**Methodology:** Taku Toyama, Takashi Ueta.

**Supervision:** Takashi Ueta.

**Writing – original draft:** Taku Toyama.

**Writing – review & editing:** Han Peng Zhou, Miki Ikeda, Taiji Hasegawa, Tomohiro Iida, Makoto Aihara, Takashi Ueta.

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
