## [Decision Letter · Decision Letter 0]

11 Oct 2024

PONE-D-24-35586Estimation of Foveal Avascular Zone Area from a B-can OCT image using Machine Learning AlgorithmsPLOS ONE

Dear Dr. Toyama,

Thank you for submitting your manuscript to PLOS ONE. After careful consideration, we feel that it has merit but does not fully meet PLOS ONE’s publication criteria as it currently stands. Therefore, we invite you to submit a revised version of the manuscript that addresses the points raised during the review process.

We look forward to receiving your revised manuscript.

Kind regards,

Tatsuya Inoue

Academic Editor

PLOS ONE

Journal Requirements:

 Author T.T. was supported by the Japan Society for the Promotion of Science (JSPS) KAKENHI (Grant Number: 22K16963). JSPS website: https://www.jsps.go.jp/english/  

Reviewers' comments:

Reviewer's Responses to Questions

**Comments to the Author**

1. Is the manuscript technically sound, and do the data support the conclusions?

Reviewer #1: Yes

Reviewer #2: Yes

2. Has the statistical analysis been performed appropriately and rigorously? 

Reviewer #1: Yes

Reviewer #2: Yes

3. Have the authors made all data underlying the findings in their manuscript fully available?

Reviewer #1: No

Reviewer #2: Yes

4. Is the manuscript presented in an intelligible fashion and written in standard English?

Reviewer #1: Yes

Reviewer #2: Yes

5. Review Comments to the Author

Reviewer #1: The manuscript states, “All relevant data are within the manuscript and its Supporting Information files.” However, I could not locate the OCT B-scans and OCTA images for the 104 subjects. Given the substantial number of images, it would be customary to use a data repository such as Figshare for sharing this kind of dataset. Could you please provide a public URL for access to these files?

As the data have been anonymized and the study was ethically approved for research use with an opt-out option, it seems that public sharing should be feasible. Could you clarify if there are any concerns regarding data availability for public access?

Comments on the Conclusion:

The conclusions may be overstated. It is unclear why the accurate inference of a single measurement from OCTA (such as FAZ) using B-scans would provide additional evidence to support the intricate interaction between neural and vascular structures within the macula. Since FAZ can be measured in a relatively short time, why is it possible to capture a B-scan but not an OCTA image in the same scenario? Are there indeed many cases where B-scans can be obtained, but OCTA imaging cannot? This point warrants further clarification.

Additionally, while the FAZ area could potentially be valuable for studies on developmental changes or age-related variations in the elderly, the more straightforward approach might involve short-time OCTA imaging of a small region focused on the FAZ, allowing for direct data acquisition. There is insufficient evidence to suggest that inference performance from low-quality B-scans, particularly in populations such as children or the elderly (who are challenging to image), is reliable.

It would be advisable to temper the conclusions and align them more closely with the specific findings of this study. Conclusions should reflect the direct implications of the data, without overextending beyond what the results demonstrate.

Reviewer #2: The authors developed machine learning models to predict FAZ metrics from OCT b scan. The subject of the manuscript can be of interest, however, I have the following comments and questions:

1. Please provide the OCT and OCTA devices used in this study

2. I think a useful statistical analysis would be Bland–Altman analysis to show 95% limit of agreement (95% LoA)

3. It is already shown that measuring FAZ in full retinal slab is more accurate that SCP or DCP. Why did you use the SCP slab

4. An important limitation of the study is including only healthy subjects as it is previously shown that in diseased eyes, even the octa built-in software is inaccurate in delineating the FAZ area (https://doi.org/10.1038/s41598-020-80058-x)

6. PLOS authors have the option to publish the peer review history of their article (what does this mean?). If published, this will include your full peer review and any attached files.

Reviewer #1: **Yes: **HIDENORI TAKAHASHI

Reviewer #2: **Yes: **Reza Mirshahi

---

## [Author Response · Author response to Decision Letter 0]

31 Oct 2024

We have uploaded our detailed point-by-point responses to the reviewers' and editor's comments as a separate file titled 'Cover letter' and 'rebuttal letter'.

---

## [Decision Letter · Decision Letter 1]

2 Dec 2024

Estimation of Foveal Avascular Zone Area from a B-can OCT image using Machine Learning Algorithms

PONE-D-24-35586R1

Dear Dr. Toyama,

We’re pleased to inform you that your manuscript has been judged scientifically suitable for publication and will be formally accepted for publication once it meets all outstanding technical requirements.

Kind regards,

Tatsuya Inoue

Academic Editor

PLOS ONE

Additional Editor Comments (optional):

Reviewers' comments:

Reviewer's Responses to Questions

**Comments to the Author**

1. If the authors have adequately addressed your comments raised in a previous round of review and you feel that this manuscript is now acceptable for publication, you may indicate that here to bypass the “Comments to the Author” section, enter your conflict of interest statement in the “Confidential to Editor” section, and submit your "Accept" recommendation.

Reviewer #1: All comments have been addressed

2. Is the manuscript technically sound, and do the data support the conclusions?

Reviewer #1: Yes

3. Has the statistical analysis been performed appropriately and rigorously? 

Reviewer #1: Yes

4. Have the authors made all data underlying the findings in their manuscript fully available?

Reviewer #1: Yes

5. Is the manuscript presented in an intelligible fashion and written in standard English?

Reviewer #1: Yes

6. Review Comments to the Author

Reviewer #1: In this revised version, the manuscript has been well improved according to the reviewers' comments. I believe the manuscript has reached a stage where there are no significant issues.

7. PLOS authors have the option to publish the peer review history of their article (what does this mean?). If published, this will include your full peer review and any attached files.

Reviewer #1: **Yes: **Hidenori Takahashi

---

## [Editor Report · Acceptance letter]

5 Dec 2024

PONE-D-24-35586R1 

PLOS ONE

Dear Dr. Toyama, 

I'm pleased to inform you that your manuscript has been deemed suitable for publication in PLOS ONE. Congratulations! Your manuscript is now being handed over to our production team.

Kind regards, 

on behalf of

Dr. Tatsuya Inoue 

Academic Editor

PLOS ONE